# Confounders in Identification and Analysis of Inflammatory Biomarkers in Cardiovascular Diseases

**DOI:** 10.3390/biom11101464

**Published:** 2021-10-05

**Authors:** Qurrat Ul Ain, Mehak Sarfraz, Gayuk Kalih Prasesti, Triwedya Indra Dewi, Neng Fisheri Kurniati

**Affiliations:** 1Department of Pharmacology and Clinical Pharmacy, School of Pharmacy, Bandung Institute of Technology, Bandung 40132, Indonesia; aineevirk.av@gmail.com (Q.U.A.); gayukkp@gmail.com (G.K.P.); 2Department of Pharmacy, Comsats University Islamabad Abbottabad Campus, Abbottabad 22060, Pakistan; mehaksarfraz6@gmail.com; 3Department of Cardiology and Vascular Medicine, Faculty of Medicine, Universitas Padjadjaran, Bandung 40124, Indonesia; t_indradewi@yahoo.co.id

**Keywords:** pro-inflammatory biomarkers, confounding factors, inflammation, chemokines, cytokines, acute-phase proteins, demographic factors, epidemiological factors, pre-analytical factors

## Abstract

Proinflammatory biomarkers have been increasingly used in epidemiologic and intervention studies over the past decades to evaluate and identify an association of systemic inflammation with cardiovascular diseases. Although there is a strong correlation between the elevated level of inflammatory biomarkers and the pathology of various cardiovascular diseases, the mechanisms of the underlying cause are unclear. Identification of pro-inflammatory biomarkers such as cytokines, chemokines, acute phase proteins, and other soluble immune factors can help in the early diagnosis of disease. The presence of certain confounding factors such as variations in age, sex, socio-economic status, body mass index, medication and other substance use, and medical illness, as well as inconsistencies in methodological practices such as sample collection, assaying, and data cleaning and transformation, may contribute to variations in results. The purpose of the review is to identify and summarize the effect of demographic factors, epidemiological factors, medication use, and analytical and pre-analytical factors with a panel of inflammatory biomarkers CRP, IL-1b, IL-6, TNFa, and the soluble TNF receptors on the concentration of these inflammatory biomarkers in serum.

## 1. Introduction

Cardiovascular diseases (CVDs) are also known as heart diseases, which refer to the following four conditions: coronary artery disease (CAD) or coronary heart disease (CHD), cerebrovascular disease, peripheral artery disease (PAD), and aortic atherosclerosis [1]. The prevalence of CVDs is increasing all over the world and is considered the most common cause of morbidity and mortality in both developed and developing countries [2]. Diets rich in salt, sugar, and lipids, reduced physical activity, use of tobacco and alcohol, and metabolic factors such as hyperlipidemia, high blood pressure, elevated body mass index, and waist–hip ratio are known causes of CVDs [3]. CVDs are sometimes asymptomatic, presented with silent ischemia and angiographic evidence of CAD without symptoms, whereas classical presentation includes typical anginal chest pain consistent with myocardial infarction (MI) and/or acute cerebral stroke [1]. There are two fundamental assumptions regarding the etiology of CVDs. They may be caused due to damage to the endothelium of blood vessels resulting in the formation of lesions, localized inflammation due to mobilized white blood cells, lipoproteins, and other substances, which lead to the development of fibrofatty atherosclerotic plaques causing stenosis of the arteries [4]. Another assumption is that CVDs may be caused when atherosclerotic plaque ruptures and forms a clot leading to arterial occlusion, thereby hindering the flow of blood and oxygen to the heart causing damage to the heart muscle and brain, resulting in MI thromboembolic stroke [4].

Not all atherosclerotic plaques are equally vulnerable; some plaques are prone to rupture and variability. Plaques have large lipid cores with the activity of inflammation and a thinner fibrous cap [5]. The vulnerability of plaque depends on size and wall stress as well as flow impact on the luminal plaque surface [6]. Despite plaque rupture, there might be the formation of a thrombus which contributes to rapid plaque progression and reduced luminal diameter. Thrombus develops at the site of vulnerable plaque when the thrombogenic lipid-rich core is exposed by rupture. Thrombosis induced at plaque rupture sites could enhance occlusion of vessels and give a complete or subtotal occlusion of coronary arteries [7]. Atherosclerosis at an early stage has an inflammatory component characterized by infiltration of leukocytes at the vascular endothelial wall. Transendothelial migration and adhesion of circulating leukocytes are thought to be important in the initiation and spread of atherosclerotic disease [8]. 

CVD is multifaceted; numerous biological pathways have been implicated, not only those limited to stress and inflammation. Various pathological processes are considered to be involved in CVD, and several evidence lines support a significant role for inflammation in the progress of diseases [9]. The role of inflammation and cytokines in CVDs is described in Figure 1.

Initiation and progression of atherosclerotic plaque are mediated by inflammatory changes that occur in endothelial cells’ (ECs) linings due to endothelial injury, abnormal lipid metabolism, and hemodynamic damage [10]. Activated endothelial cells express monocyte chemoattractant inflammatory factors such as E-selectin, P-selectin, intercellular adhesion molecule-1 (ICAM-1), and CD40 ligand, inducing the recruitment of lymphocytes and monocytes which infiltrate the wall of arteries, initiating the process of inflammation [11]. The processes are mediated by various other cells and cytokines, such as macrophages, lymphocytes (T and B cells), dendritic cells (DCs), ECs, vascular smooth muscle cells (VSMCs), interleukins (ILs), adhesion molecules, and tumor necrosis factor (TNF-α) [12]. Proinflammatory monocytes are then differentiated into macrophages that engulf lipid deposits and transform them into foam cells, and the monocytes expressing Ly6C or Gr-1 accumulate in the atherosclerotic plaque and stick to the endothelial cell lining [13]. 

The use of biomarkers for various purposes in CVD remains an imperative research area that has been explored by scientists over the years, and new developments over the past 30 years have led to additional sensitive methods of screening for early detection and diagnosis of CVDs [14]. Inflammatory biomarkers may serve to help identify patients at risk for CVD, monitor the treatment’s efficacy, and develop new pharmacological tools [15]. Due to the complexities of CVD pathogenesis, there is no single biomarker available to estimate the absolute risk of future cardiovascular events [16]. Furthermore, not all biomarkers are equal; the functions of many biomarkers overlap, some offer better prognostic information than others, and some are better suited to identify/predict the pathogenesis of particular cardiovascular events. C-reactive protein (CRP) is probably the most promising indicator for vascular inflammation established for cardiovascular events [17]. CRP is the acute-phase protein, produced in the liver during the acute phase of inflammation at the local site of infection or injury [17].

Interleukin-1b (IL-1b) belongs to the IL-1 family, which consists of three structurally related polypeptides: IL-1a, IL-1b, and IL-1 receptor antagonist (IL-1ra). IL-1b is predominantly synthesized after mononuclear phagocytes, smooth muscle cells, and endothelial cells are triggered by microbes or endogenous products, i.e., uric acid or cholesterol crystals [18]. This results in the formation of a cytosolic complex of proteins (nucleotide-binding leucine-rich repeat-containing pyrin receptors (NLRPs)) known as ‘inflammasome’, which activates caspase-1 in response to danger signals. This caspase-1 converts pro-IL-1b into the active form IL-1b [19]. Excessive production of IL-1b and an imbalance in IL-1b and IL-1ra is known to be linked with inflammation [19]. Interleukin-6 (IL-6) is a multifunctional proinflammatory cytokine that controls cellular and humoral responses and plays a vital role in tissue injury and inflammation. Various research studies have demonstrated the important roles of IL-6 in innate immune responses and adaptive immunity by activating T-helper 17 cells and inhibiting the regulatory T cells with attendant inflammation [20]. Another most important pro-inflammatory cytokine is TNF-α, which causes blood vessel dilatation, edema, and leukocyte adhesion to the epithelial cell lining that leads to blood coagulation and enhances oxidative stress at sites of inflammation [21].

Several studies have examined the inflammation associated with CVD through the measurement of a variety of analytes, such as inflammatory biomarkers, serum amyloid A [SAA], white blood cell (WBC) count, and fibrinogen [22]. However, analytical assays for biomarkers are utilized in clinical settings after carefully considering the commercial availability of these analytical assays, their sensitivity and precision measured by the coefficient of variation, stability of the biomarker, and the standardized method to carry out assays for comparison of results [22]. However, in reality, confounding factors mask an actual relationship between the treatment and its outcome, or sometimes demonstrate a false association when no real association between them exists [23]. Confounding is mostly described as the “mixing of effects” of an additional factor on the results or outcomes, which leads to a distortion of the true relationship [24]. In clinical studies, confounding occurs when a known prognostic factor differs between groups being compared in the study.

Several steps have to be passed before a potential biomarker becomes clinically significant in the diagnosis or prognosis of the disease [23]. Therefore, levels of biomarkers should be detected by an easy analytical method and a statistical association between the biomarker and the clinical state of interest should be proven, as well [25]. The confounding factors influencing the outcomes while determining levels of cytokine cardiac biomarkers can be divided into in vivo preanalytical factors, in vitro preanalytical factors, and analytical factors [26]. Therefore, factors strongly influencing the results have to be controlled by maintaining uniform conditions [26].

## 2. In vivo Preanalytical Confounders

### 2.1. Demographic Factors

#### 2.1.1. Age and Sex

Aging is associated with increased levels of circulating cytokines and proinflammatory markers [27]. According to research, aging is linked to a state of persistent low-grade inflammation and elevated serum levels of inflammatory markers such as IL-6, CRP, and TNF, a process known as “inflammaging” [28]. It is well known that CRP, the most thoroughly researched of the inflammatory biomarkers, increases with age [29]. CRP in the blood is a sensitive indicator of systemic low-grade inflammation and a strong predictor of CVDs [30]. CRP activates complement pathways and has a major role in some forms of tissue alteration, such as in cardiac infarction [31]. According to a study by Tomasik, people in their 60s and 70s have greater CRP levels than people in their 20s and 50s. When compared to the younger population, healthy elderly people have lower serum levels of CRP and pro-inflammatory cytokines [32]. 

IL-6 is one of a class of immune system regulators functioning as a pro-inflammatory cytokine that activates inflammatory pathways, such as NF-κB, MAPK, and JAK-STAT [33,34]. IL-6 is always present in the body in small amounts (<1–2 µg/mL), and its concentration varies by time of day. IL-6 levels also rise with advancing age and are related to a variety of chronic conditions [35]. IL-6 levels have risen modestly with age, whereas IL-8 levels did not. IL-8 is one of the most essential chemokines for attracting circulating neutrophils to an infection site, and it is not likely to be impacted by age [32]. CRP is an acute-phase protein produced by the liver in response to an increase in IL-6 levels. Even in healthy people and in the absence of acute infection, levels of certain cytokines, particularly IL-6 and TNF-α, rise with age [36]. Serum levels of IL-6, CRP, and TNF-R1 were greater in participants ≥65 than <65 years of age. 

Studies about sex differences in inflammatory markers in children show that initial and peak CRP was higher in girls compared with boys [37]. The 95th percentile value of CRP in the overall population was 0.95 mg/dLfor males and 1.39 mg/dLfor females and varied with age and race. For ages 25–70 yrs, the age-adjusted approximate upper reference limit (mg/dl) was CRP = age/50 for males, and CRP = age/50 + 0.6 for females [38]. CRP levels are also related to hormone levels in women and are elevated with the use of oral contraceptives or postmenopausal hormone replacement therapy [35]. High levels of CRP, between 3 and 10 mg/dL, are related to the development of CVDs [35]. CRP levels were discovered to be greater in premenopausal women than in men, but IL-6 and TNF-α levels were found to be lower in women than in men [39]. No clear explanation exists to understand how sex hormones and/or chromosomes affect the immune system [37,40].

The upper reference limit for CRP should be adjusted based on demographic characteristics such as age, sex, and race. A large multiethnic-based research study showed that black subjects had significantly higher levels of CRP levels as compared to white subjects [38]. When utilizing CRP levels to assess inflammatory disorders, clinicians should be mindful of these aspects [38]. The actual process that causes the rise with age is unknown. The reported increase in total and visceral adiposity with age, as well as falling levels of sex hormones following menopause and andropause, are proposed causes. Another process contributing to an increase in the amount of these markers could be oxidative damage caused by aging, which then triggers an inflammatory response [36]. The definition of normal ranges in the elderly should be considered [32].

#### 2.1.2. Obesity

Obesity is associated with chronic activation of the immune system and an altered gut microbiome, leading to an increased risk of chronic disease development [41]. Recent studies show that a blood level of CRP above 10 mg/dL is related to chronic conditions such as obesity and poor social conditions, because people with poor diet and a sedentary lifestyle due to poverty and illiteracy are more prone to CVDs [35]. Many systems of the body have been reported in the literature as being dysregulated in obesity, and subsequently increase the risk of chronic disease development [41]. These findings are in line with previous research that has found that obese people’s immune systems are dysregulated, resulting in a high pro-inflammatory to anti-inflammatory biomarker ratio [42]. Obese people had significantly higher CRP levels than non-obese people, regardless of whether they had metabolic syndrome (MS) [43]. Obesity is a proatherogenic condition that predisposes to CVD via its major associated risk factors such as dyslipidemia, hypertension, insulin resistance, and type 2 diabetes mellitus [44]. Besides being produced by hepatocytes, CRP is also produced by adipose tissue [44,45]. Obesity conditions lead to adipose tissue dysfunction, triggering the release of proinflammatory adipokines which can directly act on cardiovascular tissues to promote disease [46,47]. Adipokines are bioactive molecules secreted by adipose tissues that primarily work as inflammatory modulators. Obesity causes pro-inflammatory adipokines to be upregulated while anti-inflammatory adipokines are downregulated, contributing to the pathophysiology of CVDs [47]. TNF-α and IL-6 are both produced by adipose tissue, and clinical studies demonstrate that circulating levels of CRP, fibrinogen, and TNF-α are all related to body mass index (BMI). Adipose tissue is responsible for 30% of total circulating IL-6 concentrations. This is significant because IL-6 regulates CRP synthesis in the liver, and CRP can be a sign of a chronic inflammatory condition that can lead to ACS [48]. The infiltration of expanded adipose tissue by macrophages, which are responsible for both the generation of inflammatory signals and the production of cytokines such as IL-6 and TNF-α, could explain the rise in CRP in obesity [44]. To assess their predictive power in the population, we need to conduct a thorough prospective review and make comparisons with traditional risk indicators [49].

### 2.2. Epidemiological

#### 2.2.1. Arthritis

Rheumatoid Arthritis (RA) is a chronic inflammatory illness that affects around 1% of the population and is characterized by inflammation and synovitis, which leads to cartilage degradation and juxta-articular bone disintegration. In healthy men and women, elevated levels of IL-6 indicate the risk of cardiovascular events. IL-6 is highly linked to higher cardiovascular and all-cause mortality in women with RA, rather than non-fatal CVD. Elevated levels of inflammatory markers such as IL-6 and TNF-α were found in RA and correlated with high coronary calcium scores, independent of the Framingham risk score and diabetes [50]. A recent meta-analysis of 32 studies found statistically significant differences in serum CRP levels between osteoarthritis (OA) patients and healthy controls. CRP was also associated with pain and impaired physical function, but not with radiographic OA, according to the study [51].

#### 2.2.2. Diabetes

Diabetes along with higher CRP levels increases the risk of CVD in diabetic patients [52,53]. A study reported by King in 2003 showed that diabetic patients who had elevated hemoglobin A1c (HbA_1c_) levels (≥9.0%) had a significantly higher percentage of elevated CRP than people with low (<7%) HbA_1c_ levels [54]. Another prospective cohort study investigating diabetic patients in the absence of CVD showed 103 incident cases of CVD (18 myocardial infarction, 70 coronary artery bypass graft/transluminal coronary angioplasty CABG/PTCA, and 15 strokes), confirmed by medical records of five years of follow-up, were identified. Among the study population of 746 men (74.6%), case subjects had significantly higher levels of CRP [52]. Other pathogenic processes in diabetic patients, such as an increase in the development of immune complexes with changed lipoproteins, may be relevant contributors to CRP release, in addition to the advanced glycation end product–mediated cytokine release [55]. Another biomarker is IL-6, which shows a significant increase in serum levels in young type 1 diabetic patients with adequate glycometabolic control and no clinical signs of microvascular and macrovascular disorders [52].

#### 2.2.3. Autoimmune Disorders

Autoimmune illnesses are characterized by a “coordinated immunological attack” directed against self-molecules (autoantigens) that the immune system misidentifies as foreign bodies. Changes in genes that control self-tolerance pathways are important in the etiology of various disorders [56]. Patients with systemic lupus erythematosus (SLE) often display modest elevations of CRP despite raised disease activity and increased (IL-6) [57].

#### 2.2.4. Depression

Depression is one of the primary causes of disability as well as one of the leading contributors to the global burden of disease. Even though the pathophysiology is yet unknown, past research suggests that low-grade systemic inflammation may play a role in the development of depression [58]. Major depressive disorder (MDD) is associated with increased CRP compared with healthy volunteers, and the case-control difference appears higher in treatment-resistant depression [59]. Elevated levels of CRP are associated with increased risk for psychological distress and depression in the general population. Cross-sectional population studies with 5000 to 7000 participants have reported an association between CRP levels and depression [58]. Depression affects neuroendocrine pathways, which affect the etiology and progression of coronary atherosclerosis and heart disease. According to a meta-analysis of 11 cohort studies, depression, as measured by self-reported symptoms or professional psychiatric evaluation, strongly predicts a risk for initial CHD occurrences, even when other CHD risk factors are involved, such as high LDL cholesterol, low HDL cholesterol, high blood pressure, family history, diabetes, or smoking. Similarly, higher inflammatory biomarkers, particularly highly sensitive C-reactive protein (hs-CRP), have been identified as risk markers for incident CHD events, and research studies have been published explaining the cellular and molecular mechanisms by which these biomarkers aid atherosclerosis formation [60]. It is still unclear whether and to what extent elevated CRP levels are associated with psychological distress and depression in the general population [58].

#### 2.2.5. Metabolic Syndrome

Diabetes and cardiovascular events are more likely in patients with MS. The Adult treatment panel-III (ATP-III) guideline also proposes a working definition of MS, which comprises at least three of the following characteristics: abdominal obesity, raised triglycerides, lower level of high-density lipoproteins (HDL) cholesterol, high blood pressure, and high fasting glucose [61]. The Centers for Disease Control and Prevention have recommended that a CRP cut point of 3 mg/L be used to distinguish high-risk and low-risk individuals. Interrelationships between CRP, the MS, and incident cardiovascular events were examined using baseline CRP levels and median CRP levels [62]. Five years of follow-up interview study has revealed that a positive association between hs-CRP and incident MS was found only in 13% of the women, whereas no positive association was found in men [63]. Cardiovascular risk is categorized depending on the level of hs-CRP: low risk-hs-CRP <1mg/L; moderate risk-hs-CRP between 1-3mg/L, and high risk-hs-CRP >3mg/L [62,64]. hs-CRP is a protein similar to CRP, but hs-CRP is just a term for CRP assays with a much lower detection limit which are capable of producing a quantitative result in the range below 3 mg/L. As a pleiotropic cytokine, IL-6 plays an important role in various metabolic processes as an autocrine and/or paracrine action of adipocyte function. At present, accumulating evidence has demonstrated that IL-6, soluble IL-6 receptor (sIL-6r), is closely linked to metabolic disorders [61] such as type 2 diabetes [65,66]. IL-6 levels are elevated in the adipose tissues of patients with diabetes mellitus or obesity, particularly in those with MS symptoms, suggesting that IL-6 could be used as a prognostic indicator for MS and cardiovascular dysfunction. In the pathophysiology and development of MS and cardiovascular events, IL-6 may act as an early and typical marker [65,66]. Confounding factors generally increase the level of biomarkers and hence increase the chances of false prediction of CVD. In establishing a CVD diagnosis, of course, it cannot be based on an increase in one biomarker, but it is necessary to consider the clinical condition of the patient. In addition, there is a need for specific range values of biomarkers for the conditions that become confounding factors in this review.

### 2.3. Substance Use-Related Factors

The relationship between substance abuse and inflammation is evident, but it differs for the type of substance being abused. However, little research has been carried out to check if inflammation is one of the pathways that is related to substance use disorders and their clinical outcomes [67].

#### 2.3.1. Caffeine Use

Although the coffee compounds that are responsible for the suggestive protective effects are yet unclear [68], the research studies have suggested that the effect of coffee on inflammation varies in healthy individuals who consume coffee, which can be either an increase, decrease, or no effect on the concentrations of proinflammatory biomarkers [69]. According to a research study in which 33 athletes participated, 17 of athletes who took 6mg/kg of body weight of caffeine before completing a 15-km run were compared with 16 athletes in the placebo group. The measurement of oxidative stress markers in blood samples showed that exercise and caffeine consumption by subjects under study resulted in significantly higher concentrations of the biomarkers IL-6 and IL-10 in plasma levels [70].

#### 2.3.2. Alcohol

Alcohol intake, even in moderate amounts, causes complex changes in blood biochemistry, involving changes in many biomarkers for cardiometabolic risk [71]. Few research studies have been conducted to evaluate the association of alcohol with cardiac biomarkers, cardiac wall stretch, and systemic low-grade inflammation [72]. Research studies mostly focused on populations with relatively moderate alcohol intake, except for one study which shows a link between heavy drinking and heart failure in men with underlying myocardial ischemia [72]. According to measurements made of hsCRP, the most extreme drinking pattern shows the highest levels of all CRP biomarkers in comparison to people who don’t drink. This result postulates that heavy alcohol drinking affects cardiac structure and function adversely, in a way that may not be caused by atherosclerosis [73]. In contrast to previous studies that rely on self-reported alcohol consumption, there is another study investigating the relationship between alcohol intake and cardiac biomarkers in men. The study suggests that men with alcohol consumption have a higher concentration of biomarkers that result in a higher risk for cardiac remodeling, leading to atrial fibrillations [74].

#### 2.3.3. Smoking

The mechanism behind smoking and its link with cardiac damage leading to incident heart failure independent of CAD is not well understood [75]. Several large-scale and well-controlled studies have shown that there are complex mechanisms behind the effect of cigarette smoke on the lungs and circulating proinflammatory cytokines. There is a graded relationship between the amount smoked over a lifetime and an increased level of IL-6 and CRP inflammatory markers, in addition to the effect of current smoking status [76].

A study including 35 teenagers aged 10–18 measuring the levels of IL-6 and CRP demonstrated that CPR was higher in smokers, and that IL-6 did not correlate with smoker status or number of cigarettes smoked per day [75].

### 2.4. Medication-Related Factors

#### 2.4.1. Antidepressants

A key mechanism behind the significant adverse effects of antidepressant medication is low-grade systemic inflammation that particularly increases CVD risk. Most research studies have shown that the use of antidepressant drugs has been associated with a high risk of cardiovascular incidents [77]. Antidepressants are associated with a higher risk of elevated CRP in users of tricyclic antidepressant (TCA) medication, but not in selective serotonin reuptake inhibitors (SSRIs) users. According to the Whitehall cohort, the use of antidepressants was associated with elevated levels of systemic inflammatory biomarker CRP independently from the symptoms of mental illness and cardiovascular comorbidity [78].

#### 2.4.2. Nonsteroidal Anti-Inflammatory Drugs (NSAIDs) and Anticoagulants

NSAIDs have analgesic, anti-inflammatory, and antipyretic therapeutic properties. These drugs are the most commonly used over-the-counter drugs as well as prescription drugs for various clinical conditions, i.e., pain, RA, OA, musculoskeletal disorders, and other comorbid conditions [79]. An increase in blood pressure and the development of congestive heart failure are also widely recognized by the use of these drugs. Rofecoxib is associated with a higher risk of acute myocardial infarction in clinical trials because of the potential cardiotoxicity of selective cyclooxygenase-2 inhibitors [80]. According to a meta-analysis of controlled trials, it is evident that NSAIDs do not affect the CRP level. However, the use of nonselective NSAID naproxen significantly lowers the CRP level, whereas the cyclooxygenase 2-selective NSAID lumiracoxib significantly increases the CRP level, influencing cardiovascular complications [81]. Anticoagulants such as heparin, given to patients before blood sampling, may change the levels of biomarkers in blood. This has been supported by a research study in which several hundred proteins were quantified after an acute myocardial infarction before heparin administration and after heparin administration in 500 individuals. It was found that 25 of 653 identified plasma proteins showed a changed in their concentrations after heparin administration, whereas 14 of the proteins were significantly changed in patients before heparin treatment [82].

#### 2.4.3. Statins and Anti-Hypertensive Medications

Baseline troponin is an independent biomarker of myocardial infarction or death from CHD, and its concentration is lowered by statin therapy. A study revealed that Pravastatin lowered the concentration of troponin by 13% and doubled the number of men for whom troponin fell, which showed them as having the lowest risk for future coronary events [83]. Moreover, it has been postulated that angiotensin-converting enzyme (ACE) inhibitors are therapeutically beneficial due to their anti-inflammatory activity and reduction in local or systemic expression of IL-6. Concomitant increases in plaque angiotensin II expression28 may drive IL-6 expression. IL-6 is co-localized with ACE within atherosclerotic plaques, suggesting a possible local role of inflammation in the initiation and progression of atherosclerosis [84]. A research study in which the control group received aspirin after undergoing surgery and the observation group received ACE inhibitors captopril and valsartan after surgery, showed that the observation group had significantly lower levels of IL-6, hs-CRP, and TNF-α than the control group at one, four, and eight weeks after treatment [85]. The in vivo preanalytical confounders that affects the levels of inflammatory biomarkers in cardiovascular diseases are mentioned in Table 1.

## 3. In vitro Preanalytical Confounding Factors

Common laboratory variables include in vitro preanalytical confounding factors, such as sample collection methods, handling storage, freeze-thaw cycles, and type of specimen, i.e., plasma versus serum cytokines, as well as analytical factors related to assay methodology and standardization, may affect the concentration level of cytokines which may lead disparities seen among similar types of clinical studies [86]. Multicentric studies show more variations in predictive values of an inflammatory marker because strict adherence to protocols for sample collection may be more achievable in homogenous monocenter studies, but is more challenging in multicenter studies. Temperature and time between collection and performance of analytical assay are critical for sample integrity and stability [87]. However, most of the research studies have been performed by collecting blood samples within the laboratory, but don’t show operational conditions of large-scale field surveys because transportation of samples to the analytical laboratory can be a potential variable [88].

### 3.1. Incubation, Storage, and Collection

Methods for proper sample collection and assaying vary in the research literature. Technical details such as how the sample (blood) is drawn, incubated, and stored can significantly affect the results of immunoassay, especially if the sample collected from different patient groups is not treated the same way. Research studies have demonstrated that multiple freeze–thawing cycles of samples results in protein degradation, which leads to artifactual peaks in mass spectrometry analyses [89]. The sample collection procedure may influence the values of biomarkers because the stability of cytokines may vary, and sometimes cytokines may be taken up by leukocytes after blood sample collection [90]. One research study has found that the high sensitivity and stability of CRP in multiple freeze–thaw cycles of the assay may make it a particularly useful biomarker of CVDs. A delay up to 6 h in specimen processing and storage temperature did not affect levels of CRP [88]. According to one study, the level of TNF-α decreases 50% when there is a delay in plasma sample processing during the separation of plasma from cells [86]. Production of cytokine in collected whole blood becomes apparent in two hours following sample collection, and if the whole blood sample is left at room temperature, the level of IL-6 decreases significantly after four hours [91]. Under unseparated conditions, the half-life of IL-6 is short because of degradation that occurs during storage [92].

### 3.2. Diurnal Variability

Another well-known source of preanalytical variability is the diurnal, monthly, and seasonal variation shown by many biomarkers [93]. Research literature reports that circadian rhythm plays an important role in triggering cardiovascular events by the implication of physiological rhythms that show peak activity at particular times of the day or night. The inflammatory functions associated with a higher incidence of cardiovascular events vary over 24 hours [94]. Circadian time structure has been shown to affect every biological function tested in human beings, including some cytokines. Rudnicka et al. conducted a cross-sectional study of seasonal and diurnal fluctuations in fibrinogen, hs-CRP, fibrin D-dimer, tissue plasminogen activator antigen, and von Willebrand factor in a large number of men and women aged 45 years. These researchers have shown that this diurnal rhythm is associated with the variation in these biomarkers. However, for all biomarkers, the amplitude of the diurnal variation was greater than the seasonal variation [95]. Normally, the majority of clinical studies are conducted during daytime hours, when the subject is awake. To reduce the discrepancies between the studies carried out in different laboratories, as well as between animal and human subjects, the variation due to diurnal rhythm on the intrinsic properties of the cardiovascular system should be considered during the design of in vivo experimental studies. Validation of the biomarkers is important to demonstrate their reliability, stability, and lack of variability, and to standardize the methodology of their measurement [96].

### 3.3. Centrifugation and Heat Denaturation

Lack of access to centrifugation due to sample collection in particular regions and remote locations exposes the specimen to variable temperatures before the separation of serum/plasma [87]. Tirumalai et al. pointed out that high centrifugal force may compromise the integrity of the membrane and allow high molecular weight components, such as albumin, to pass through, when using non-diluted plasma and when the ultrafiltration was conducted under nondenaturing solvent conditions [97]. Additionally, it seems that low-speed centrifugation by using diluted serum or plasma under denaturing conditions is an important variable in performing plasma/serum centrifugal ultrafiltration to remove larger proteins, along with low molecular weight (LMW) protein/peptides [97]. To avoid pre-analytical variations that occur due to differences in sample handling conditions, ideal pre-analytical conditions should be maintained uniformly. Therefore, the stability of biomarkers should be ensured when fluctuating pre-analytical conditions such as temperature and time taken for centrifugation, in particular, are encountered [88].

### 3.4. Epitope Masking and/or Assay Specific Variability

Cardiovascular biomarkers are mostly heterogeneous peptides and proteins in nature that affect the accuracy of immunoassay by cross-reacting with the antibodies used in immunoassay systems, affecting the accuracy of the measurement. As a result, it is not surprising that there are large systematic differences between the circulating levels of biomarkers measured by immunoassay methods [98]. Assays having acceptable analytical imprecision and high sensitivity with a low detection limit (LoD) of about 1 pg/tube or even lower are required to measure circulating levels of cardiac biomarkers because they are present in very low concentrations in tissues and body fluids of healthy subjects [99]. Moreover, to reduce inter-assay variability, assay manuals suggest that blood samples of CRP are tested within the same run, and samples should be tested in duplicate [89]. The in vitro preanalytical confounders that affects the levels of inflammatory biomarkers in cardiovascular diseases are mentioned in Table 2.

## 4. Analytical Confounding Factors

### 4.1. Within-Subject Correlation

Engelberger et al. assessed the variation in levels of biomarkers within-subject and between-subject in patients with biological PAD. The study concluded that due to large within-subject variability, single biomarker hs-CRP measurements are not very useful both in PAD patients and healthy subjects [100]. According to the study conducted by Hijazi et al., the average changes in biomarker levels over 2 months were small when cardiac troponin I, cardiac troponin T, and N-terminal pro-B-type natriuretic peptide levels were measured in patients with stable atrial fibrillation by using high-sensitivity immunoassays after 2 months. Therefore, repeated measurement of cardiac biomarkers provides some significant prognostic value for mortality but not for stroke, when combined with clinical risk factors and baseline levels of the biomarkers [101]. In a cross-sectional study, Abramson et al. examined inflammatory markers including CRP and TNF-α to check BP variability within-subject during the daytime, nighttime, and 24-hour periods. It was shown that CRP has positive associations with nighttime and 24-hour systolic BP variability, and TNF-α was not associated with systolic BP variability during any of the periods, whereas, with regards to diastolic BP variability, CRP was positively associated with diastolic BP variability during all periods and TNF-α was also positively associated with daytime diastolic BP variability. Therefore, it is evident that within-subject variability exists between inflammatory biomarkers and BP in healthy normotensive adults [102].

### 4.2. Reproducibility Issues

Reproducibility issues in serial measurements also occur due to variation in analytical methods and day-to-day inter- and intra-subject variations [103]. It is important to know about these variations to estimate the value of a single time point’s concentration of biomarkers to clearly define reference values for comparison of a significant change in the values over time due to any confounding reason [104]. The retrospective research on data analysis of the association between the elevated inflammatory markers and short-term ACS outcomes made reproducibility issues evident by showing that data collected in electronic records can affect that results if, for patients admitted multiple times to the hospital, the dataset for each of that patient’s hospital stays is included in the study [105].

### 4.3. Selection Bias

Selection bias occurs when the selected subjects for a particular study are not representative of the overall population, and therefore the outcome of exposure will be associated with a significant bias, which eventually leads to a result that varies from what you can obtain if you enroll the whole targeted population [106]. Various research studies have shown that patient selection based on discharge diagnosis leads to a selection bias [105]. There are instances when there are many analyses performed but only some of them with the “best” results are reported, and this results in selective analysis reporting bias [107]. But the presence of bias in the results does not mean that levels of inflammatory cardiac biomarkers have no association with cardiovascular outcomes. Therefore, it is difficult to differentiate whether the underlying effect of selection bias is small or null, and whether genuine heterogeneity exists. However, there are also several research studies of cardiovascular biomarkers in the literature that did not show any evidence of biases [108]. One more aspect of bias in determining levels of cardiovascular biomarkers is selective reporting biases, that are more commonly seen to exaggerate the proposed associations of these biomarkers with cardiovascular events. According to the systematic evaluation of 56 meta-analyses of emerging cardiovascular biomarkers to determine the effect of selective bias, 29 meta-analyses (52%) were shown to have statistically significant heterogeneous results [108]. Tzoulaki et al. found strong evidence of the inflated effect of biomarkers because the largest studies, which have been expected to produce the most stable estimates, consistently showed smaller effects, whereas meta-analyses of many single studies showed positive results as compared to the results of the largest studies [107,108]. This revealed that small studies with “negative” results remain unpublished, or that their results are distorted during analysis and reporting to seem more prominent. Therefore, selective reporting bias of positive findings has been a major problem in clinical evaluation and biomarker research in particular [109]. Some retrospective studies of hospitalized patients to identify CRP cardiac biomarkers associated with inflammation have shown the significant risk of selection bias because the indication for measurements is at the discretion of the treating physician [110].

### 4.4. Data Analysis Concerns

Research studies vary in methods for handling and cleaning data. Transparent data management techniques are required to avoid non-detects and high-value outliers, replication, and reproducibility issues in datasets. Moreover, different researchers could theoretically reproduce distinct findings within a single dataset by using different data management techniques [111]. The analytical confounders that affects the levels of inflammatory biomarkers in cardiovascular diseases are mentioned in Table 3.

## 5. Conclusions

Preanalytic and analytical variability is an important aspect that affects the quality of the laboratory-determined concentrations of cardiac biomarkers. To correctly evaluate cardiac biomarkers’ results, physicians, especially cardiologists and laboratory specialists, should be aware of all the aspects which could affect the quality of laboratory results. They must remember that these confounders are often overlooked, which could be a significant source of bias.

## Figures and Tables

**Figure 1 biomolecules-11-01464-f001:**
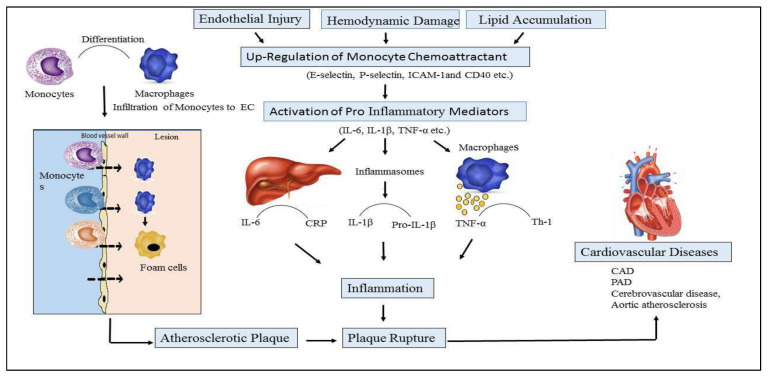
Role of inflammation and cytokines in the pathophysiology of cardiovascular disease (CVDs). Endothelial cells (EC), Interleukin-6 (IL-6), Interleukin-1b (IL-1b), Tumor Necrotic Factor-α ((TNF-α), C-reactive protein (CRP), Cardiac artery disease (CAD), Peripheral Artery Disease (PAD).

**Table 1 biomolecules-11-01464-t001:** In vivo Preanalytical Confounders in Identification and Analysis of Inflammatory Biomarkers in Cardiovascular Diseases.

Confounders	Summary	Reference
Demographic factors		
*Age and Sex*	Aging increased serum levels of IL-6, CRP, and TNF-α; women have CRP levels higher than men	[27,28,32,35,36], [38]
*Obesity*	Obese people had significantly higher levels of CRP, TNF-α, and IL-6 than non-obese people.	[43,44]
Epidemiological factors		
*Arthritis*	In RA conditions, there is an increase in IL-6, TNF-α, and CRP levels in OA	[51]
*Diabetes*	In diabetes, there is an increase in IL-6 and CRP levels	[52,53]
*Autoimmune Disorders*	In autoimmune disorders, there is an increase in IL-6 and CRP levels	[57]
*Depression*	In depression, there is an increase in both hs-CRP and CRP levels	[59,60]
*Metabolic Syndrome*	In autoimmune disorders, there is an increase in IL-6 and CRP levels	
Substance use-related factors		
*Caffeine use*	Caffeine consumption resulted in significantly higher concentrations of biomarkers IL-6 and IL-10 in plasma levels	[70]
*Alcohol*	Alcohol consumption resulted in increased in hs-CRP	[72,73]
*Smoking*	Alcohol consumption resulted in increased in CRP and IL-6	[75,76]
Medication-related factors		
*Antidepressants*	Antidepressants are associated with a higher risk of elevated CRP in users of tricyclic antidepressant (TCA) medication	[78]
*NSAIDS*	Cyclooxygenase 2-selective NSAID lumiracoxib significantly increases the CRP level influencing cardiovascular complications	[81]
*Statins and anti-hypertensive medications*	Statin therapy lowered troponin levels; captopril and valsartan lowered IL-6, hs-CRP, and TNF-α	[82,84]

**Table 2 biomolecules-11-01464-t002:** In vitro Pre-analytical Confounders in Identification and Analysis of Inflammatory Biomarkers in Cardiovascular Diseases.

Confounders	Summary	Reference
Incubation, Storage, and Collection	Delay up to 6 h in specimen processing and storage temperature did not affect levels of CRP, but TNF-α decreased 50%	[20,89]
Diurnal Variability	No research study available demonstrating the effect of diurnal variation in natriuretic peptides	[93]
Centrifugation and Heat Denaturation	Stability of biomarkers should be ensured, such as temperature and time taken for centrifugation	[88]
Epitope Masking and/or Assay Specific Variability	Assays having acceptable analytical imprecision and high sensitivity with a low detection limit (LoD) of about 1 pg/tube	[99]

**Table 3 biomolecules-11-01464-t003:** Analytical Confounders in Identification and Analysis of Inflammatory Biomarkers in Cardiovascular Diseases.

Confounders	Summary	Reference
Within-Subject Correlation	CRP has positive associations with nighttime and 24-hour systolic BP variability	[102]
Reproducibility Issues	Serial measurements also occur due to variation in analytical methods and day-to-day inter- and intra-subject variations	[103]
Selection Bias	Research studies have shown the significant risk of selection bias in CRP measurement	[110]
Data Analysis Concerns	Transparent data management techniques are required to avoid non-detects and high-value outlier replication, and reproducibility issues	[111]

## Data Availability

No new data were created or analyzed in this study.

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
