# Peer review of "Confounders in Identification and Analysis of Inflammatory Biomarkers in Cardiovascular Diseases"

_biomolecules, 2021, doi:10.3390/biom11101464_

Round 1

Reviewer 1 Report

This review manuscript entitled “Confounders in Identification and Analysis of Inflammatory Biomarkers in Cardiovascular Diseases” first described current pro-inflammatory biomarkers associated with cardiovascular disease followed by summarizing the confounding effect of demographic factors, epidemiological factors, medication use, and analytical / pre-analytical factors on the concentration of a panel of inflammatory biomarkers including CRP, IL-1b, IL-6, IL-18, TNF-a, or the soluble TNF receptors in patient’s serum.

Overall, this is an up-to-date and comprehensive review presenting potential confounders in identification and analysis of inflammatory biomarkers in cardiovascular diseases. However, a few minor typos need to correct before publication.

  1. Once the abbreviation had been defined, the authors should use the abbreviated term thereafter (no need to re-define again later). For example, cardiovascular diseases (CVDs), C-reactive protein (CRP), interleukin-6 (IL-6), tumor necrosis factor (TNF), or N-terminal pro-B-type natriuretic peptide (NT-proBNP).
  2. Please define CVS in line 60.
  3. In line 61, please rephrase “but not limited e stress and”.
  4. In line 99 or 104, IL-1 should be specified as IL-1b.
  5. In line 111, coronary artery disease (CAD) should be defined.
  6. In line 205 and 210, please correct TNF- to TNF-α.
  7. In line 390, please define ACE, then remove definition of ACE in line 396.
  8. In line 488, please correct “Lack (not lack) of access”.

Reviewer 2 Report

To the authors:

  1. General comments:

The review entitled “Confounders in Identification and Analysis of Inflammatory Biomarkers in Cardiovascular Diseases” describes the association of promising blood biomarkers in cardiovascular diseases and their correlation with different analytical and biological confounders. The topic is interesting and according to my research this topic has not been covered before. However, the manuscript has significant flaws that the authors should address and because of that I consider should be rejected.

  1. Specific comments for revision: b) major.
    1. The manuscript should be re-organized and authors must avoid repetitions, review contradictory phrases and review the English language as some phrases are not understandable. I suggest that the introduction should describe the inflammatory related biomarkers and then their relation within the CVDs, then how they are affected by the analytical and biological confounders. During the text seems to be focused in CRP and IL-6, TNF, but then IL-8, NT-proBNP, hsTnT, homocysteine, fibrinogen, miR-1, miR-423-5p, galectin-3, ST2, cardiac tropnins… appear with no justification.
    2. Please carefully review that you have plenty of words which start with capital letters such as line 23 for Inflammatory.
    3. Please review the abbreviations, there are plenty of words missing abbreviations and plenty that are abbreviated several times. Abbreviation is always carried out by writing the definition and then in parenthesis the abbreviation, please change line 39.
  2. Minor comments:
    1. Line 14. “different types of cancers.” However cancer has not been included.
    2. Lines 50-53. Please add reference(s).
    3. Line 58-59. Review the sentence.
    4. Line 60. Define post-MI
    5. Lines 58-71. Please add reference(s).
    6. Lines 73-74. Please add reference(s).
    7. Line 99. This is the first time that IL appears so define first.
    8. Line 111. How IL-6R has a dominant role in CAD
    9. Line 111. Define CAD
    10. Lines 111-114. Phrases seem mixed and repeated between CAD and atherosclerosis
    11. Line 116. Which analytes? Such as…
    12. Line 116. Which assays? Such as…
    13. In vivo and in vitro, both are two separated words that always go in italics
    14. Line 151. Which inflammatory pathways
    15. Lines160-161. Please review the phrase, low levels compared with what? Add also which pro-inflammatory cytokines. Delete or re-phrase repeated information.
    16. Line 175. Add something about race.
    17. Line 176-178. The sentence is too basic
    18. Line 187. Blood level of what?
    19. Line 187-188. Why this is related with poor social conditions
    20. Obesity section. Be consistent, use diabetes or diabetes mellitus
    21. Lines 211-212. Repeated idea
    22. Line 223. Repeated: IL-6 indicate the risk of cardiovascular events
    23. Line 232. Add reference
    24. Define OA, hs-CRP, CHD, CAD, HbA1c
    25. Line 235. If CVD has been defined later only use this abbreviation
    26. Please add a line explaining the difference between CRP and hs-CRP
    27. Line 239-240. Repeated phrase
    28. Define CABG/PTCA
    29. Line 244. Add the % of men. Is this representative?
    30. Lines 259-260. Review sentence
    31. Lines 272. Which other CHD risk factors? Such as…
    32. Lines 288-291. Review sentence
    33. Lines302-303. Repeated sentence
    34. Lines 308-309. Review sentence
    35. Line 330. Do not use contractions
    36. Line 352-354. The sentence has not relation with smoking
    37. Line 366-368. Please review the sentence.
    38. Line 390. Define ACE
    39. Line 398-399. Please review the sentence.
    40. Line 423. Odd sentence, all biomarkers are measured by mass spectrometry? Not spectroscopy
    41. Lines 431-433. Contradictory sentence about multiple freeze-thaw cycles
    42. Lines 436-439. Why suddenly you introduce mi-RNAs
    43. Line 450. Define PSG
    44. Lines 457-459. Review the sentence
    45. Lines 483-485. Please review the sentence
    46. Line 496. Please define LMW
    47. Line 510-512. Review the sencence
    48. Line 536-537. Contradictory sentence to before

Reviewer 3 Report

In general the review is adequate, I like to make the following comments:

1. There are some errors detected in the PDF text.

2. The bibliography is not current. The authors shoud include more recent articles.

Citation 3 is a  Guideline published in 2008, when there are other recent ones such as

European Heart Journal, Volume 42, Issue 14, 7 April 2021, Pages 1289–1367, https://doi.org/10.1093/eurheartj/ehaa575

I suggest to include in the introduction text and its respective citations:

 Inflammatory cells such as neutrophils, followed by monocytes and macrophages, rapidly infiltrate the injured myocardium with abundant proinflammatory cytokine secretions that may cause additional damage to the myocardium. These inflammatory processes have been identified as key mediators of reperfusion injury in ST-segment– elevation myocardial infarction (STEMI).

Bibliography:

Bochaton T, Lassus J, Paccalet A, Derimay F, Rioufol G, Prieur C, Bonnefoy-Cudraz E, Crola Da Silva C, Bernelin H, Amaz C, et al. Association of myocardial hemorrhage and persistent microvascular obstruction with circulating inflammatory biomarkers in STEMI patients. PLoS One. 2021;16:e0245684. doi: 10.1371/journal.pone.0245684

Frangogiannis NG. The inflammatory response in myocardial injury, repair, and remodelling. Nat Rev Cardiol. 2014;11:255–265. doi: 10.1038/ nrcardio.2014.28

And “In vitro preanalytical confounding factors” section should include:

Beck HC, Jensen LO, Gils C, Ilondo AMM, Frydland M, Hassager C, Møller-Helgestad OK, Møller JE, Rasmussen LM. Proteomic Discovery and Validation of the Confounding Effect of Heparin Administration on the Analysis of Candidate Cardiovascular Biomarkers. Clin Chem. 2018 Oct;64(10):1474-1484. doi: 10.1373/clinchem.2017.282665. Epub 2018 Aug 16. PMID: 30115630.

3. Include a table or tables listing the confounding factors to make it easier to follow the text.

Round 2

Reviewer 2 Report

To the authors:

  1. General comments:

Thanks to the authors for addressing the suggestions of the previous revision. I consider the work has been significantly improved. However, I found some minor changes that the authors should change before acceptance of the manuscript:

  1. Specific comments for revision: b) minor.
    1. Line 13-14. Please review the sentence. It is not understood.
    2. Line 94. ILs has not been defined
    3. Line 96. Ly6C has not been defined
    4. Line 196. In vitro goes in italics
    5. Line 320,386, 799. Use the abbreviation for cardiovascular disease
    6. Line 390, . Use the abbreviation for acute coronary syndrome
    7. Lines 408-410. Please review the sentence. It is not understood at all, and check also which are the cytokines the authors refer to?
    8. Line 481. Use the abbreviation for interleukin
    9. Line 560-561. Please review the sentence. It is not understood.
    10. Line 575-577. Please add what this information means? Higher concentrations of IL-6 and IL-10 points to inflammation caused by exercise and coffee consumption?
    11. Line 640-641. Delete this sentence, this has been said before
    12. Line 650. Use the abbreviation for RA and OA
    13. Line 737-738. Please review the sentence. It is not understood.
    14. Line 742. Use the abbreviation for CHD
    15. Line 755, 903. In vitro goes in italics
    16. Line 796. Cytokines in capital letter?
    17. Line 798. “The” in capital letter?
    18. Lines805-814. I don’t agree with this part at all. Fasting affects lipids, but it affects proteins and peptides? This review is based on these molecules. Moreover, long sleep is not related with fasting time
    19. Line 918-919. I don’t agree with this phrase as well. Is not that large-scale studies being not feasible, it is more difficult to achieve but if the center has the equipment to follow SOPs and it is well-planned can be achieved.
    20. Line 982. Please define in the text BP
    21. Line 1055., 1056, 1057. Please check the capital letters for Bias, Cardiac and Retrospective
    22. Line 1062. Please check detects”
